# Intranasal Administration of Apelin-13 Ameliorates Cognitive Deficit in Streptozotocin-Induced Alzheimer’s Disease Model via Enhancement of Nrf2-HO1 Pathways

**DOI:** 10.3390/brainsci14050488

**Published:** 2024-05-11

**Authors:** Hai Lu, Ming Chen, Cuiqing Zhu

**Affiliations:** 1State Key Laboratory of Medical Neurobiology, Institutes of Brain Science, MOE Frontier Center for Brain Science, Fudan University, Shanghai 200032, China; 15111010013@fudan.edu.cn (H.L.); ming_chen@fudan.edu.cn (M.C.); 2College of Clinical Medicine, Jining Medical University, Jining 272067, China

**Keywords:** Alzheimer’s disease, apelin-13, intranasal administration, oxidative stress, Nrf2-HO-1 signaling pathway

## Abstract

Background: The discovery of novel diagnostic methods and therapies for Alzheimer’s disease (AD) faces significant challenges. Previous research has shed light on the neuroprotective properties of Apelin-13 in neurodegenerative disorders. However, elucidating the mechanism underlying its efficacy in combating AD-related nerve injury is imperative. In this study, we aimed to investigate Apelin-13’s mechanism of action in an in vivo model of AD induced by streptozocin (STZ). Methods: We utilized an STZ-induced nerve injury model of AD in mice to investigate the effects of Apelin-13 administration. Apelin-13 was administered intranasally, and cognitive impairment was assessed using standardized behavioral tests, primarily, behavioral assessment, histological analysis, and biochemical assays, in order to evaluate synaptic plasticity and oxidative stress signaling pathways. Results: Our findings indicate that intranasal administration of Apelin-13 ameliorated cognitive impairment in the STZ-induced AD model. Furthermore, we observed that this effect was potentially mediated by the enhancement of synaptic plasticity and the attenuation of oxidative stress signaling pathways. Conclusions: The results of this study suggest that intranasal administration of Apelin-13 holds promise as a therapeutic strategy for preventing neurodegenerative diseases such as AD. By improving synaptic plasticity and mitigating oxidative stress, Apelin-13 may offer a novel approach to neuroprotection in AD and related conditions.

## 1. Introduction

Alzheimer’s disease (AD) presents a formidable challenge in terms of diagnosis and treatment, spurring intensified research efforts. The apelin system has recently garnered attention for its potential involvement in AD pathogenesis, as evidenced by an expanding body of literature [1,2].

Apelin, a biologically active peptide hormone, originates from fat cells [3]. It exists in three active forms, comprising either 13, 17, or 36 amino acids, derived from a 77-amino-acid prepropeptide precursor by an angiotensin-converting enzyme [4,5]. Notably, Apelin-13 demonstrates significantly greater biological potency than Apelin-36 [4]. The apelin system has demonstrated therapeutic potential across various acute and chronic neurological conditions [6]. Studies have shown its efficacy in alleviating acute brain injuries, such as subarachnoid hemorrhage, traumatic brain injury, and ischemic stroke. Additionally, it has been found to have therapeutic effects on chronic neurodegenerative disease models, involving the regulation of neurotrophic factors, neuroendocrine signaling, oxidative stress, neuroinflammation, neuronal apoptosis, and autophagy [1,7,8]. Moreover, apelin exhibits neuroprotective properties by mitigating oxidative damage in neurons. In vivo experiments have shown that apelin’s ability to combat reactive oxygen species (ROS) and free radicals is closely related to its neuroprotective effects against neurodegenerative diseases [9].

Apelin-13 and its receptor APJ are widely distributed in the hippocampus and other brain tissues. The intracerebroventricular injection of Apelin-13 improved stress-induced memory function decline in rats [10], indicating that apelin/APJ signaling may be involved in cognitive ability [11,12]. Moreover, apelin has been implicated in enhancing the function of various factors, including GLP-1, eNO, and ACE2, thereby promoting synaptic plasticity and improving learning and memory. Additionally, apelin exerts a neuroprotective effect by attenuating inflammatory responses through the BDNF signaling pathway [13]. Furthermore, apelin can modulate amyloid-beta (Aβ) metabolism by reducing amyloid precursor protein (APP) levels and inhibiting β-secretase activity, leading to decreased Aβ production and increased Aβ clearance mediated by ABCA1 and NEP. Additionally, apelin may mitigate tau protein phosphorylation and accumulation. Studies have reported that apelin can prevent neurodegeneration by reducing levels of inflammatory mediators, particularly TNF-α and IL-1β, and inhibit neuronal apoptosis by modulating the balance of anti-apoptotic and pro-apoptotic factors [14]. These findings underscore the multifaceted neuroprotective effects of apelin, suggesting its potential as a therapeutic target in Alzheimer’s disease intervention.

While oxidative stress is recognized as a crucial mechanism underlying Apelin-13’s anti-AD effects, the specific oxidative stress pathway through which apelin mitigates AD-related neurological damage requires further investigation. To assess the effectiveness of Apelin-13, we employed STZ-induced AD mice as an in vivo model system and included donepezil, a standard AD therapeutic, as a positive control. By analyzing parameters such as cognitive function, synaptic plasticity, oxidative stress markers, and signaling pathways, we aimed to clarify the therapeutic potential of Apelin-13 and its mechanism of action. Employing a range of methodological approaches, our study aims to elucidate the precise molecular pathways through which Apelin-13 exerts its neuroprotective effects against AD-associated oxidative stress. The novelty and scientific significance of our study lie in the exploration of intranasal Apelin-13 administration as a potential therapeutic strategy for AD, offering insights into its neuroprotective effects and underlying mechanisms.

## 2. Materials and Methods

### 2.1. Experimental Animals

Male adult C57BL/6J mice, aged between 8 and 12 weeks, were accommodated in a controlled environment with a 12 h light/dark cycle, ensuring regulated temperature and humidity levels. They had ad libitum access to food and water throughout the study period. All experimental procedures adhered to the ethical guidelines established by Fudan University and international standards for animal research. Approval for the animal study protocol was granted by the Animal Care and Use Committee of the Shanghai Medical College of Fudan University (No. 20170223-098). Every effort was taken to minimize animal discomfort and to optimize the utilization of animals in the study.

### 2.2. Experimental Protocol

Mice were categorized in a random manner into different groups (10 animals each). Group 1 served as the control group and received ICV saline (0.9%). Group 2 was the STZ-induced AD group, receiving 3 mg/kg STZ (ICV) based on previous studies [15]. Group 3 was the positive control group, treated with donepezil at a dose of 5 mg/kg/day via oral administration. Groups 4 and 5 received a single dose of 3 mg/kg STZ (ICV). Two days later, Group 4 received nasal administration of Apelin-13 at a dose of 1 mg/kg, while Group 5 received nasal administration of Apelin-13 at a dose of 0.2 mg/kg. Behavioral testing was performed sequentially after the injected program ended. Following the final behavioral assessment, the animals were euthanized and sampled. Specifically, three rats from each group were selected for electrophysiological tests, while another three rats were utilized for biochemical analyses and Western blot tests, depending on the specific objectives of the experiment.

### 2.3. Drug Administration

For nasal administration, the Apelin-13 (0.2 mg/kg, 1 mg/kg) solution (10 μL) was pipetted bilaterally on the rhinarium, i.e., the glabrous skin around the nostrils, and allowed to diffuse in the squamous epithelium. After the drip, the mouse was fixed for 5–10 s to ensure the liquid was fully inhaled. Donepezil (Sigma-Aldrich, St. Louis, MO, USA) were dissolved in 0.9% saline. A freshly prepared solution of donepezil was administered orally daily at a dose of 5 mg/kg [16]. ML385 (Medchem Express, South Brunswick, NJ, USA) is a specific inhibitor of Nrf2. Before each administration of Apelin-13, the mice received an intraperitoneal injection of 30 mg/kg ML385 [17], dissolved in saline containing 50% PEG300, with a 30 min interval. Zn(II)-protoporphyrin IX (ZnPP, Medchem Express, South Brunswick, NJ, USA) is an inhibitor of HO-1. Mice were injected intraperitoneally with 25 mg/kg ZnPP [18] (dissolved in saline) 30 min before each administration of Apelin-13.

### 2.4. Stereotaxic Surgery

Mice were anesthetized via a mixture of ketamine and xylazine and, then, secured onto a stereotaxic apparatus (Stoelting Apparatus, Wood Dale, IL, USA). To induce an STZ-AD mouse model, STZ (3 mg/kg, ICV) was bilaterally injected into the lateral ventricles. Control mice underwent the same procedure but received a vehicle injection of the citrate buffer 0.05 mol/L, pH 4.5. STZ was freshly diluted in the citrate buffer prior to injection. The injections were administered using a Hamilton syringe (model 705, Hamilton Company, Giarmata, Romania) with the following coordinates relative to the bregma: AP −0.5 mm, ML ±1.1 mm, and DV −2.8 mm. Each lateral ventricle was infused with a total infusion of 1.5 μL of STZ or citrate buffer at a rate of 0.5 μL/min. 

### 2.5. Morris Water Maze Test

The Morris water maze (MWM) test was conducted following previously established protocols [19]. Each group consisted of eight male mice. Throughout the training phase, the platform was located in the same position (one of the four pool quadrants). Each mouse was placed into the pool facing the wall at one of the four starting positions, and its movement was tracked using a digital tracking system. If a mouse reached the platform, it was immediately removed from the water. In case a mouse failed to locate the platform within 60 s during the training trials, it was gently guided to the platform or given an additional 15 s on the platform before being withdrawn from the pool. The animals underwent four daily trials over five consecutive days, with approximately 30 min intervals between trials. On the sixth day following the final training session, a probe trial was conducted. During this trial, the platform was removed from the pool, and each mouse was placed in the pool facing the wall from the diagonally opposite side of the platform. The mice were allowed to swim freely for 2 min while their movements were recorded using EthoVision software (Version XT6.1), after which they were removed from the pool.

### 2.6. Y-Maze Test

The Y-maze utilized in this study was constructed from opaque and non-reflective materials to ensure consistent testing conditions. Each arm of the maze contained distinct visual cues to facilitate spatial orientation for the mice. During testing, mice were individually placed at the end of one arm with their heads facing the central area of the maze. Two prominent markers were positioned opposite each other within the maze to serve as visual cues. Mice were then allowed to freely explore the maze for a duration of 8 min. Video recordings of the mouse’s movement trajectory were captured from above the maze using Anymaze software (Version 4.99). A mouse was considered to have entered an arm when all four of its limbs entered that arm. The three arms of the maze were designated as a, b, and c. If a mouse sequentially entered three different arms, it was deemed to have exhibited spontaneous alternation behavior. The correct rate of spontaneous alternation was calculated by subtracting 2 from the total number of completed alternations and dividing by the total number of arm entries, multiplied by 100. Between each test session, the inner surface of the maze was cleaned with 75% alcohol to eliminate residual olfactory cues and maintain consistent testing conditions for subsequent trials.

### 2.7. In Vitro Electrophysiology

Electrophysiology procedures were conducted following previously established methods [20]. Upon quick dissection, brains were immersed in artificial cerebrospinal fluid (ACSF) with the following composition: 125 mM NaCl, 2.5 mM KCl, 2 mM CaCl_2_, 1 mM MgCl_2_, 25 mM NaHCO_3_, 1.25 mM NaH_2_PO_4_, and 10 mM glucose, and then, saturated with 95% O_2_ and 5% CO_2_ at approximately 0 °C. Coronal brain slices (300 μm thick) were prepared using a vibratome and transferred to a chamber maintained at 31 °C. Slices were allowed to incubate for at least 1 h before commencing patch-clamp recording. Neurons targeted for whole-cell patch-clamp recording were accessed using glass electrodes with a resistance ranging from 5 to 8 MΩ when filled with the patch pipette solution. The internal solution of the electrode comprised 115 mM CsMeSO_3_, 10 mM HEPES, 2.5 mM MgCl_2_, 20 mM CsCl_2_, 0.6 mM EGTA, 10 mM Na phosphocreatine, 0.4 mM Na-GTP, and 4 mM Mg-ATP. Each experimental group included three male mice for long-term potentiation (LTP) recording. Excitatory postsynaptic currents (EPSCs) were recorded in CA1 neurons, with a concentrated stimulating electrode placed in the Schaffer collaterals. LTP was induced using the 3× theta burst stimulation protocol (TBS), involving four pulses at 100 Hz repeated with 200 ms inter-burst intervals. The average EPSC amplitudes 30 min after LTP induction in each group were compared to assess potential differences in LTP magnitude among the groups. Data acquisition and analysis were performed using pClamp10.7 and Clampfit 10.7 (Axopatch 700B, Molecular Devices, San Jose, CA, USA). 

### 2.8. The Detection of Oxidative Stress Levels

Hippocampal tissues were collected from each group for cell disruption using a sonicator (Huxi Company, Shanghai, China), followed by centrifugation at 3500 r/min for 10 min. Supernatants from groups 1–5 were analyzed to measure the activities of the antioxidant enzymes of superoxide dismutase (SOD), glutathione peroxidase (GSH-Px), catalase (CAT), and the concentration of alondialdehyde (MDA). The SOD (A001-1-2), MDA (A00-1-2), CAT (A007-1-1), and GSH-Px (A005-1-2) test kits were acquired from the Nanjing Jiancheng Biological Research Institute, China. Specifically, SOD and CAT were measured using Coomassie brilliant blue and hydroxylamine methods, respectively. MDA levels were determined through the ammonium molybdate method, while GSH-Px levels were analyzed using the thiobarbituric acid colorimetric method.

### 2.9. Western Blot Analysis

Upon treatment, tissues were rinsed and, then, immersed in PBS before being subjected to centrifugation. Cell lysis was conducted at 4 °C by vigorously shaking for 15 min in RIPA buffer comprising 150 mM NaCl, 1% NP-40, 0.5% sodium deoxycholate, 0.1% SDS, 50 mM Tris–HCl (pH 7.4), 50 mM β-glycerol phosphate, 20 mM NaF, 20 mM EGTA, 1 mM DTT, and 1 mM Na_3_VO_4_, along with protease inhibitors. Subsequent to centrifugation at 15,000 rpm for 15 min, the supernatant was isolated and stored at −70 °C until further use. Protein concentration was determined utilizing the Bradford method, and the lysates were boiled for 5 min. Denatured proteins were resolved via sodium dodecyl sulfate–polyacrylamide gel electrophoresis on 8% or 10% polyacrylamide gels and, then, transferred onto PVDF membranes (Millipore, Billerica, MA, USA). Following overnight blocking at 4 °C in 5% BSA in Tris-buffered saline/Tween [containing 0.05% (*v*/*v*) Tween 20], the membranes were incubated with specific antibodies at dilutions of 1:2000 [phospho-ERK1/2 (Thr202/Tyr204) (Abcam, Cambridge, UK, ab278538), ERK (Abcam, ab32537), anti-Nrf2 (MedChemExpress, South Brunswick, NJ, USA, YA895), and anti-HO-1 (Millipore, 374087)]. A subsequent incubation was performed using a horseradish peroxidase-conjugated secondary anti-rabbit IgG antibody (dilutions of 1:5000, Signalway Antibody, L3012). Blots were developed using the ECL Western blotting detection reagent (Santa Cruz Biotechnology, Dallas, TX, USA).

### 2.10. Statistical Analysis

Numerical data are presented as mean ± SEM. Analysis of offline data was conducted using Clampfit software (Version 10.5, Axon Instruments, San Jose, CA, USA) and GraphPad Prism 6 (GraphPad Software, La Jolla, CA, USA). Statistical significance was assessed by ANOVA followed by Bonferroni post-tests for multiple comparisons among groups. For electrophysiological tests, ‘*n*’ represents the number of cells, with each cell group in every experiment sourced from a minimum of four animals. The threshold for statistical significance was set at *p* < 0.05.

## 3. Results

### 3.1. Intranasal Administration of Apelin-13 Improves Cognitive Impairment in STZ-Induced Animal Model of AD Mice

The Morris water maze and Y-maze tests were conducted after 30 days of intranasal Apelin-13 treatment in an STZ-induced animal model of AD to determine whether intranasal administration of Apelin-13 improves spatial learning and memory. We designed parallel positive drug control experiments using donepezil as an effective positive control. First, we examined the open-field behavior of the different treatment groups, and the experimental results showed that there were no differences in the total movement distance of the different drug treatment groups within the 15 min test time (Figure 1B, *n* = 7–10, one-way ANOVA, *p* > 0.05). There were no differences between the different treatment groups in terms of movement time or proportion of edge movement (Figure 1C,D, *n* = 7–10, one-way ANOVA, *p* > 0.05). Trace records (Figure 1E) showed that the swimming trajectory of the STZ-induced AD mice in the target quadrant was shorter than that of the control mice and Apelin-13-treated mice. There was no significant difference in swimming speed between the groups (Figure 1F). The STZ-induced AD mice required more time to find the target quadrant and performed fewer platform crossings. Furthermore, both parameters significantly improved after Apelin-13 treatment in the STZ-induced animal model of AD (Figure 1G, H, *n* = 7–10, one-way ANOVA, *p* < 0.05). 

In the Y-maze test (Figure 1I), there was no significant difference in the total number of entries between the different groups (Figure 1J, *n* = 7–10, one-way ANOVA, *p* > 0.05). No significant difference in the alternation ratio between the control and high-Apelin-13 treatment group existed. However, the alternation rate of the high-Apelin-13 treatment group was higher than that of the STZ-induced AD group (Figure 1K, *n* = 7–10, one-way ANOVA, *p* < 0.05). In summary, our data indicate that the intranasal administration of Apelin-13 effectively improves cognitive impairment in an STZ-induced animal model of AD. 

### 3.2. Intranasal Administration of Apelin-13 Restores LTP in CA1 Neurons of STZ-Induced AD Mice

We measured evoked EPSC and LTP in hippocampal slices to investigate the effect of intranasal Apelin-13 administration on synaptic plasticity in the CA1 region of the hippocampus. Figure 2A shows representative EPSC traces before and after TBS stimulation in the four groups of mice used in this study. The ratio of LTP was significantly reduced in the STZ-induced animal model of AD compared to that in control mice. The depression of LTP 30 min after TBS in the STZ-induced animal model of AD treated with Apelin-13 confirmed that Apelin-13 significantly enhances the magnitude of LTP (Figure 2B,C, *n* = 8, one-way ANOVA, *p* < 0.05). These data confirm that intranasal administration of Apelin-13 can rescue impaired LTP in APP/PS1 mice.

### 3.3. Intranasal Administration of Apelin-13 Reduces the Oxidative Stress of the Hippocampus in STZ-Induced AD Mice

Many in vivo experiments have shown that Apelin-13 can improve oxidative stress-related indicators in animal models of AD [13]. Therefore, we used biochemical methods to observe the performance of a series of oxidative-stress-related systems in the different treatment groups. After the behavioral tests were performed for each group, we measured the activity of superoxide dismutase (SOD), glutathione peroxidase (GSH-Px), catalase (CAT), and the level of malondialdehyde (MDA) in the hippocampal tissue. The results showed that the activities of SOD, GSH, and CAT in the STZ-induced AD mouse model were significantly lower than those in the control group, and the level of MDA was increased, which was consistent with previous reports [21,22]. Donepezil treatment improved these oxidative stress indicators in an animal model of STZ-induced AD. High and low doses of Apelin-13 also improved these oxidative stress indicators in the STZ-induced AD mice (Figure 3, *n* = 3, one-way ANOVA, *p* < 0.05). These results indicate that intranasal administration of Apelin-13 can reduce oxidative stress by improving the free radical scavenging capacity in vivo.

### 3.4. Effect of Intranasal Administration of Apelin-13 on the Expression of ERK-Nrf2-HO-1 in STZ-Induced AD Mice

After confirming that the intranasal administration of apelin could improve oxidative stress factors in the hippocampus, we further verified the signaling pathway through which apelin exerts its anti-oxidative stress effect. Previous studies have shown that Aβ25-35 treatment can inhibit the expression of Nrf2 and HO-1 in SH-SY5Y cells [20]. These findings indicate that Apelin-13 may exert anti-neural-injury effects in AD cell models by activating the Nrf2-HO-1 pathway. 

First, we examined whether ERK, the classic upstream molecule of the Nrf2-HO-1 signaling pathway, changes in hippocampal tissue. The results showed that STZ caused an increase in the expression of phosphorylated ERK and that this increase was not altered by the positive control drugs donepezil and Apelin-13 (Figure 4A, *n* = 3, one-way ANOVA, *p* < 0.05). These results suggest that phosphorylated ERK protein may not be involved in the anti-AD damage mechanism of Apelin-13 administered intranasally. 

We then observed the expression of Nrf2 and HO-1 proteins in the different treatment groups. The results showed that both proteins decreased in the STZ-induced animal model of AD. This effect can be altered by donepezil and high-dose Apelin-13. There was no difference in Nrf2 and HO-1 expression between the donepezil and high-dose Apelin-13 treatment groups and the control group (Figure 4B,C, *n* = 3, one-way ANOVA, *p* < 0.05). These results indicate that the Nrf2-HO-1 signaling pathway may be one of the mechanisms of action of Apelin-13 against STZ-induced nerve injury. 

### 3.5. Inhibition of Nrf2 and HO-1 Pathways Attenuates Cognitive Benefits of Intranasal Apelin-13 Administration in STZ-Induced AD Mice

Furthermore, to elucidate the involvement of the Nrf2-HO-1 pathway in the behavioral effects of Apelin-13, we investigated the involvement of the Nrf2-HO-1 pathway in mediating the cognitive effects of Apelin-13. Mice were treated with the Nrf2 (ML385, 30 mg/kg) or HO-1 inhibitors (ZnPP, 25 mg/kg) before receiving high-dose Apelin-13, followed by an assessment of spatial learning and memory in the Morris water maze and Y-maze tests (Figure 5A). 

The Morris water maze test revealed that mice treated with high-dose Apelin-13 exhibited significantly improved spatial learning compared to the control group. However, pretreatment with the Nrf2 inhibitor or the HO-1 inhibitor attenuated these improvements, as evidenced by the shorter time in the target quadrant and the fewer platform crossings compared to the Apelin-13-treated group (Figure 5B–D, *n* = 8, one-way ANOVA, *p* < 0.05). 

In the Y-maze test, mice administered high-dose Apelin-13 demonstrated enhanced alternation behavior, indicative of improved spatial working memory, compared to the control group. Conversely, pretreatment with the Nrf2 inhibitor or the HO-1 inhibitor reversed this effect, resulting in a significant decrease in alternation behavior compared to the Apelin-13-treated group (Figure 6B,C, *n* = 8, one-way ANOVA, *p* < 0.05). The results revealed that pretreatment with the Nrf2 inhibitor or the HO-1 inhibitor abolished the cognitive improvement induced by Apelin-13, suggesting that the Nrf2-HO-1 pathway may play a crucial role in mediating the behavioral effects of Apelin-13 in AD mice.

## 4. Discussion

The main findings of the present study were that (1) the intranasal administration of Apelin-13 improves cognitive impairment in an STZ-induced animal model of AD, and (2) the anti-STZ-induced-nerve-injury effect of this administration may be achieved by improving synaptic plasticity and anti-oxidative stress signaling pathways.

An innovation of this study is a new exploration of Apelin-13 drug delivery methods. Previous studies have shown that Apelin-13, as a short peptide, cannot be administered intraperitoneally or subcutaneously, similar to conventional drugs, and mostly can only be administered intraventricularly. Many studies have reported that insulin administered through the nasal cavity can be directly absorbed by the nasal mucosa without passing through the peripheral blood circulation and can enter the brain through the blood–brain barrier to treat neurodegenerative diseases such as AD and cognitive impairment [23,24,25]. Therefore, we used the intranasal administration of Apelin-13 in mice to explore the feasibility of its drug effects.

After a series of behavioral tests in the open field, Y-maze, and water maze, we found that the intranasal administration of Apelin-13 can improve cognitive dysfunction in mice with STZ-induced nerve injury. Our behavioral results are consistent with the latest research reports [13,26,27], although in these studies, Apelin-13 was administrated using the previous ventricular cannula method. Our experimental results confirmed that nasal administration of Apelin-13 treatment can improve the behavioral performance of animal models of nerve injury, indicating that intranasal administration of Apelin-13 is effective and feasible. In future studies, we will explore the effects of the drug at multiple concentrations and time points.

After confirming the validity of the behavioral results, we focused on specific brain regions. It has been reported that apelin and its receptors are expressed in the whole brain, but the hippocampus is one of the regions with relatively high expression [28,29,30]; this region is also strongly associated with AD. Therefore, we focused on changes that occurred in the hippocampus.

We first observed the long-term potentiation of hippocampal synaptic plasticity, which is closely related to learning and memory functions, in different treatment groups. The results showed that LTP was significantly impaired in the STZ-induced nerve injury group, and high-dose Apelin-13 treatment could improve the LTP impairment caused by STZ-induced nerve injury. 

Many in vivo experiments have shown that Apelin-13 can improve oxidative stress-related indicators in STZ-induced nerve injury models [13]. Our experiments showed that the activities of SOD, GSH, and CAT in STZ-induced nerve injury mice were significantly lower than those in the control group, and the activity of MDA was increased, which is consistent with previous reports. These results indicate that the free radical scavenging ability of mice with nerve injuries was decreased, leading to oxidative damage. High and low doses of Apelin-13 also improved oxidative stress indicators in mice with nerve injuries to a certain extent.

Next, we explored the mechanism by which Apelin-13 improves STZ-induced nerve injury. Studies have reported that Apelin-13 exerts neuroprotective effects through anti-inflammatory factors, BDNF/TrkB, PGC-1α/PPARγ, and other signaling pathways [8,13,31,32,33], and that GLP-1 can improve the learning and memory functions of STZ-injured rats and inhibit tau protein hyperphosphorylation [34,35,36]. Apelin-13 protects neurons by strengthening autophagy and attenuating early-stage postspinal cord injury apoptosis in vitro [8]. In this study, we investigated whether the ERK-Nrf2-HO-1 pathway, which is related to oxidative stress, plays a role in the Apelin-13 anti-AD cell model of nerve injury. Certainly, the role of ERK in AD pathology is complex and multifaceted. Previous studies have reported conflicting findings regarding the involvement of ERK phosphorylation in the development of AD [37]. Activation of the Ras/ERK pathway has been implicated in predisposing to AD pathogenesis [38], while inhibition of this pathway has been suggested to have a protective effect [39]. Our results suggest that phosphorylated ERK protein may not be directly involved in the anti-AD mechanism of intranasally administered Apelin-13. While our study observed alterations in phosphorylated ERK levels following STZ-induced nerve injury and subsequent Apelin-13 treatment, the exact role of ERK signaling in mediating the therapeutic effects of Apelin-13 remains unclear. Further investigations are warranted to fully elucidate the specific molecular pathways through which Apelin-13 exerts its neuroprotective effects and to determine the interplay between ERK signaling and these pathways in the context of AD.

A recent report confirmed that Aβ25-35 treatment can inhibit the expression of Nrf2 and HO-1 in SH-SY5Y cells [40]. Some studies have reported that Apelin-13 reduces the number of Aβ plaques in the hippocampus [41], and this needs to be further investigated in our future studies to clarify whether the anti-AD effect of Apelin-13 is related to the reduction in Aβ plaques. Our study demonstrated that STZ treatment increases phosphorylated ERK levels and decreases Nrf2 and HO-1 expression. However, the Nrf2 and HO-1 proteins were increased compared with the STZ group after treatment with Apelin-13. The additional behavioral results further corroborate the notion that Apelin-13 exerts an ameliorative effect on cognitive impairment through the Nrf2-HO1 pathway. These findings provide additional support for the involvement of the Nrf2-HO1 pathway in mediating the cognitive-enhancing effects of Apelin-13. Our data support the finding that Apelin-13 regulates antioxidant signaling to ameliorate inflammation-induced AD symptoms, and it remains to be thoroughly investigated whether others, such as inflammatory factors, neurotransmitter signaling, etc., are also involved and whether this pathway still plays a role in Apelin-13’s treatment of other causes of AD.

Initially, we explored the protective mechanism of the intranasal administration of Apelin-13 against STZ-induced nerve injury. Importantly, it was confirmed for the first time that the administration of Apelin-13 through the nose can produce effective drug effects, similar to the effects of previous intraventricular administration techniques. Overall, these behavioral findings provide important insights into the mechanisms underlying the cognitive-enhancing effects of Apelin-13 and underscore the therapeutic potential of targeting the Nrf2-HO1 pathway for the treatment of Alzheimer’s disease. This study has several limitations that need to be considered. First, the AD model used in this study was relatively homogeneous, and further investigation is needed to confirm whether similar mechanisms operate in other animal models of AD. Second, the upstream and downstream molecules regulated by Apelin-13 in the Nrf2-HO1 pathway need to be explored in depth to provide a stronger basis for target development. These limitations highlight areas for future research aimed at expanding our understanding of the therapeutic potential of Apelin-13 in AD.

## 5. Conclusions

In summary, our study demonstrates that intranasal administration of Apelin-13 effectively improves cognitive impairment in an STZ-induced AD model by enhancing synaptic plasticity and modulating the Nrf2-HO1 pathway to reduce oxidative stress. These findings underscore the therapeutic potential of Apelin-13 in AD treatment, highlighting its innovative drug delivery method and providing valuable insights into its neuroprotective mechanisms.

## Figures and Tables

**Figure 1 brainsci-14-00488-f001:**
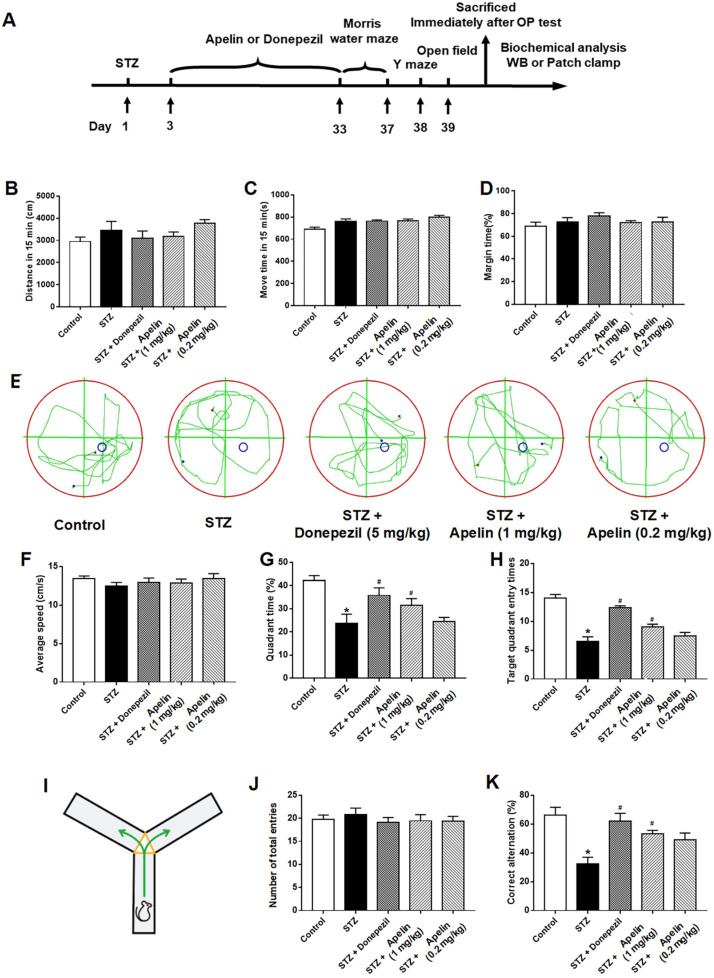
Effect of intranasal administration of Apelin-13 on cognitive impairment in STZ-induced animal model of AD mice. (**A**) Experimental timeline. (**B**) Distance of mouse movement during the 15 min test period for different groups (one-way ANOVA, *p* > 0.05). (**C**) Movement time during the 15 min test period for different groups (one-way ANOVA, *p* > 0.05). (**D**) Percentage of time spent in the margin area for different groups (one-way ANOVA, *p* > 0.05). (**E**) The swimming trajectory of mice during the probe test, Blue and red dots represent the starting and ending points of the mouse trajectories, respectively. Blue circles indicate the location of the platform. (**F**) Average speed of different groups (one-way ANOVA, *p* > 0.05). (**G**) Quadrant time of different groups (one-way ANOVA, * *p* < 0.05 compared to the control group, # *p* < 0.05 compared to the STZ treatment group). (**H**) Target quadrant entry times of different groups (one-way ANOVA, * *p* < 0.05 compared to the control group, # *p* < 0.05 compared to the STZ treatment group). (**I**) Y-maze test. (**J**) Number of arm entrances of different groups (one-way ANOVA, *p* > 0.05). (**K**) Alternation ratio of different groups (one-way ANOVA, * *p* < 0.05 compared to control group, # *p* < 0.05 compared to STZ treatment group). Data are shown as the mean ± s.e.m.

**Figure 2 brainsci-14-00488-f002:**
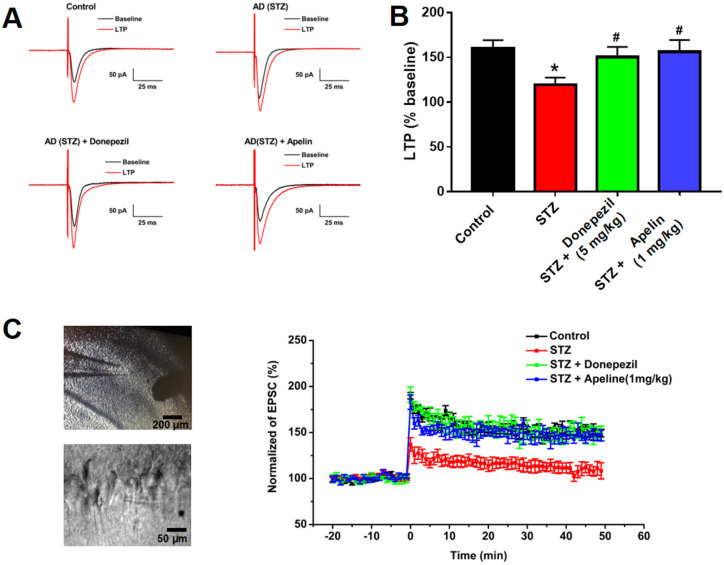
Effect of intranasal administration of Apelin-13 on LTP in CA1 neurons of STZ-induced AD mice. (**A**) Typical traces in different treatment groups. (**B**) Bar graphs showing changes in LTP for different treatment groups (*n* = 6, one-way ANOVA, * *p* < 0.05 compared to control, # *p* < 0.05 compared to STZ treatment group). (**C**) Left: Typical recording cell images (upper panel: bar = 200 μm, lower panel: bar = 50 μm). Right: Time course of the LTP in different treatment groups (*n* = 6). Data are shown as the mean ± s.e.m.

**Figure 3 brainsci-14-00488-f003:**
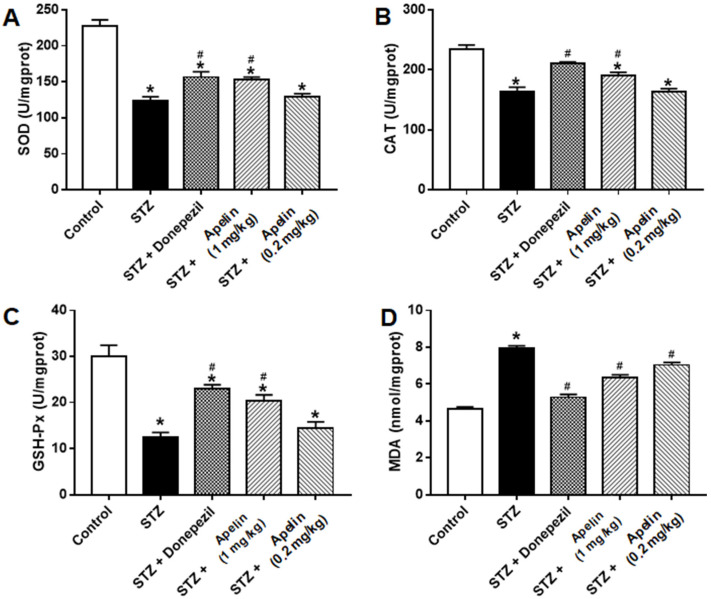
Effect of intranasal administration of Apelin-13 on STZ-induced oxidative stress in the hippocampus. (**A**) Specific activity (U/mg) of SOD in each treatment group (*n* = 3, one-way ANOVA, * *p* < 0.05 compared to control, # *p* < 0.05 compared to STZ treatment group). (**B**) Specific activity (U/mg) of CAT in each treatment group (*n* = 3, one-way ANOVA, * *p* < 0.05 compared to control, # *p* < 0.05 compared to STZ treatment group). (**C**) Specific activity (U/mg) of GSH in each treatment group (*n* = 3, one-way ANOVA, * *p* < 0.05 compared to control, # *p* < 0.05 compared to STZ treatment group). (**D**) The level (nmol/mg) of MDA in each treatment group (*n* = 3, one-way ANOVA, * *p* < 0.05 compared to control, # *p* < 0.05 compared to STZ treatment group). Data are shown as the mean ± s.e.m.

**Figure 4 brainsci-14-00488-f004:**
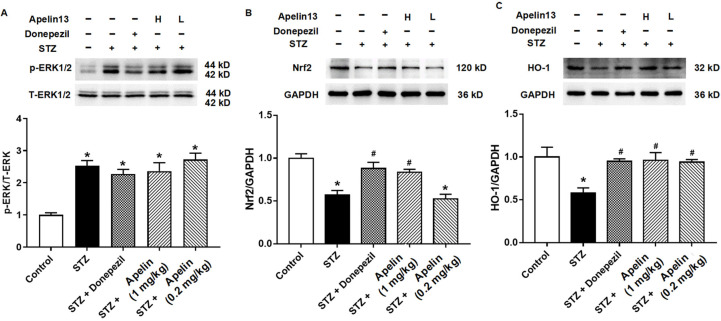
Effect of intranasal administration of Apelin-13 on the expression of ERK-Nrf2-HO-1 in STZ-induced AD mice. (**A**) Representative image of immunoblots and densitometric analysis of changes in levels of ERK family proteins in different treatment groups (*n* = 3, one-way ANOVA, * *p* < 0.05 compared to control). (**B**) Representative image of immunoblots and densitometric analysis of changes in levels of Nrf2 proteins in different treatment groups (*n* = 3, one-way ANOVA, * *p* < 0.05 compared to control, # *p* < 0.05 compared to STZ treatment group). (**C**) Representative image of immunoblots and densitometric analysis of changes in levels of HO-1 proteins in different treatment groups (*n* = 3, one-way ANOVA, * *p* < 0.05 compared to control, # *p* < 0.05 compared to STZ treatment group). Data are shown as the mean ± s.e.m.

**Figure 5 brainsci-14-00488-f005:**
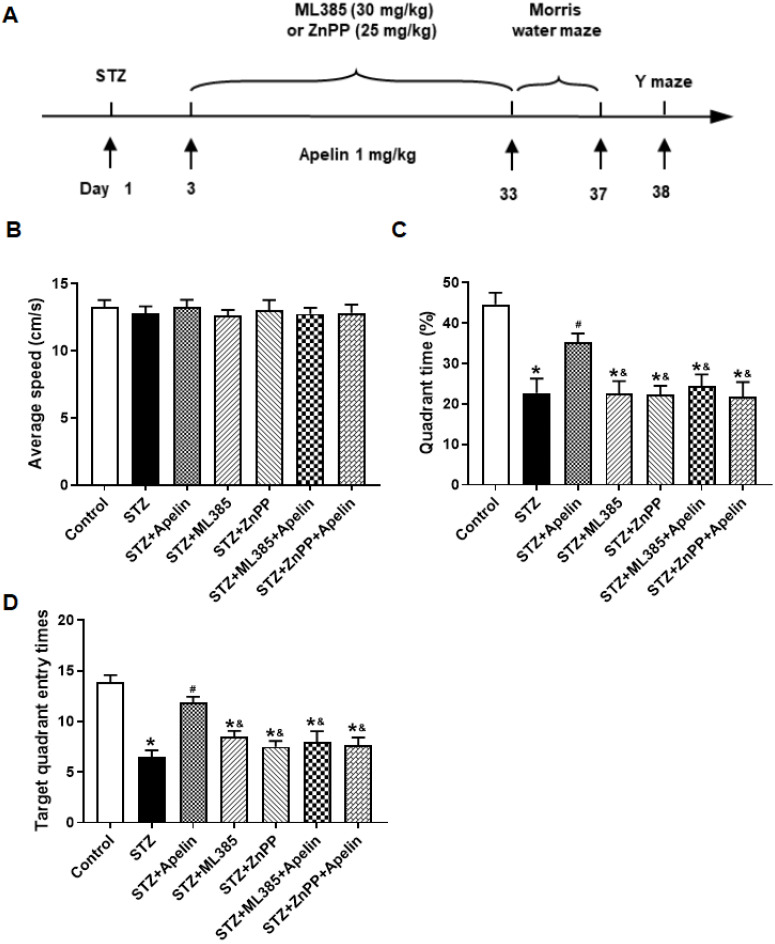
Effect of Nrf2-HO-1 pathway blockade on Apelin-13-mediated improvement of Morris maze performance in STZ-induced AD mice. (**A**) Experimental timeline. (**B**) Average speed of different groups (one-way ANOVA, *p* > 0.05). (**C**) Quadrant time of different groups (*n* = 8, one-way ANOVA, * *p* < 0.05 compared to the control group, # *p* < 0.05 compared to STZ treatment group, & *p* < 0.05 compared to STZ+Apelin-13 treatment group). (**D**) Target quadrant entry times in different groups (*n* = 8, one-way ANOVA, * *p* < 0.05 compared to the control group, # *p* < 0.05 compared to STZ treatment group, & *p* < 0.05 compared to STZ+Apelin-13 treatment group). Data are shown as the mean ± s.e.m.

**Figure 6 brainsci-14-00488-f006:**
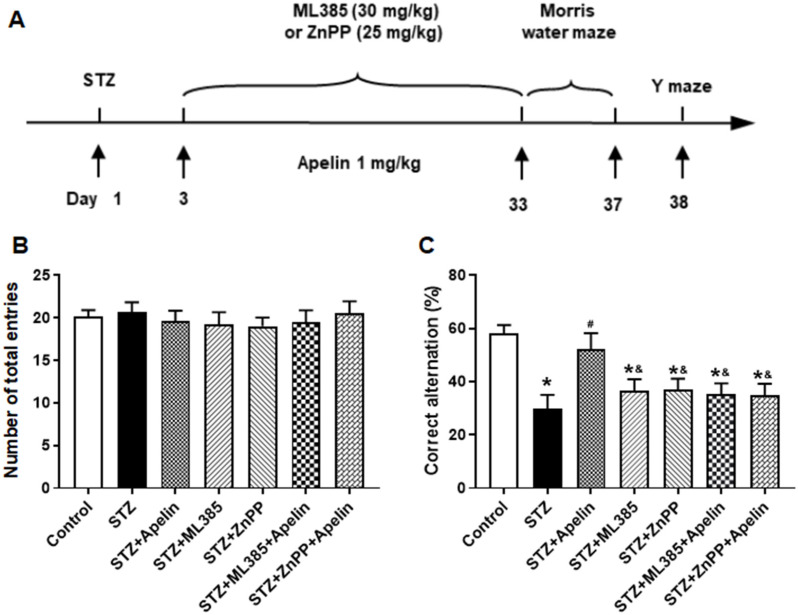
Effect of Nrf2-HO-1 pathway blockade on Apelin-13-mediated improvement of Y-maze performance in STZ-induced animal model of AD mice. (**A**) Experimental timeline. (**B**) Number of arm entrances of different groups (one-way ANOVA, *p* > 0.05). (**C**) Alternation ratio of different groups (*n* = 8, one-way ANOVA, * *p* < 0.05 compared to the control group, # *p* < 0.05 compared to STZ treatment group, & *p* < 0.05 compared to STZ+Apelin-13 treatment group). Data are shown as the mean ± s.e.m.

## Data Availability

The datasets used and analyzed in this study are available from the corresponding author upon request. The data are not publicly available due to institutional copyright policy.

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
