# Peer review of "Intranasal Administration of Apelin-13 Ameliorates Cognitive Deficit in Streptozotocin-Induced Alzheimer’s Disease Model via Enhancement of Nrf2-HO1 Pathways"

_brainsci, 2024, doi:10.3390/brainsci14050488_

Round 1
Reviewer 1 Report
Comments and Suggestions for Authors
The article “Intranasal administration of Apelin-13 ameliorates cognitive deficit in STZ-induced Alzheimer’s disease model via enhancement of Nrf2-HO1 pathways” addresses the protective properties of Apelin 13. The findings are interesting, but I have some suggestions to improve manuscript:
a) Please proofread the article, some minor corrections need to be made such as “ANOVS” to “ANOVA”.
b) In methods section, please
b.1) Indicate ethical approval number in methods.
b.2) Specify the “high” and “low” apelin doses.
b.3) Indicate primary and secondary antibody dilutions and brand;
c) Please explain the different number of mice used in the manuscript/experiments.
d) ERK is involved in multiple processes in AD pathology, this should be discussed.
e) Can authors please elaborate on the sentence “These results indicate that phosphorylated ERK protein may be involved in the anti-AD damage mechanism of intranasal administration of Apelin-13”?
Comments on the Quality of English Language
Please proofread the article, some minor corrections need to be made such as “ANOVS” to “ANOVA”.
Author Response
Reviewers' comments: 
Reviewer #1:
Comments and Suggestions for Authors
The article “Intranasal administration of Apelin-13 ameliorates cognitive deficit in STZ-induced Alzheimer’s disease model via enhancement of Nrf2-HO1 pathways” addresses the protective properties of Apelin 13. The findings are interesting, but I have some suggestions to improve manuscript:
Comments:
- a) Please proofread the article, some minor corrections need to be made such as “ANOVS” to “ANOVA”.
Reply:
Thank you for your attention to detail. The article has been thoroughly proofread, and minor corrections such as "ANOVS" have been rectified to "ANOVA."
Comments:
- b) In methods section, please
b.1) Indicate ethical approval number in methods.
Reply:
In the methods section, the ethical approval number has been included to ensure transparency and compliance with ethical standards. Thank you for your suggestion.
Comments:
b.2) Specify the “high” and “low” apelin doses.
Reply:
In response to your feedback, the "high" and "low" doses of apelin have been specified in the methods and results section to provide clarity on the experimental protocol. Thank you for bringing this to our attention.
Comments:
b.3) Indicate primary and secondary antibody dilutions and brand;
Reply:
In accordance with your suggestion, the methods section has been updated to include information about the primary and secondary antibody dilutions, as well as the brand. This addition aims to enhance the transparency and reproducibility of the experimental procedures. Thank you for your valuable input.
Comments:
- c) Please explain the different number of mice used in the manuscript/experiments.
Reply:
Thank you for your comments. The manuscript has been revised to address the discrepancy in the number of mice used across different experiments. We adhered to the principle of minimizing the use of animals in accordance with animal ethics guidelines. After conducting behavioral tests, we allocated different groups of animals to various experiments, including electrophysiology, biochemistry, and Western blot assays. This allocation resulted in variations in the number of animals used for each experiment.
Comments:
- d) ERK is involved in multiple processes in AD pathology, this should be discussed.
Reply:
Thank you for highlighting this important aspect for discussion. Certainly, the revised manuscript now includes a discussion on the involvement of ERK in multiple processes in AD pathology.
Comments:
- e) Can authors please elaborate on the sentence “These results indicate that phosphorylated ERK protein may be involved in the anti-AD damage mechanism of intranasal administration of Apelin-13”?
Reply:
We are very sorry for the error in this expression, our experimental results suggest that ERK is not involved in the process, so the sentence was changed to the following:
“These results suggest that phosphorylated ERK protein may not be involved in the an-ti-AD damage mechanism of Apelin-13 administered intranasally.”
Comments on the Quality of English Language
Comments:
Please proofread the article, some minor corrections need to be made such as “ANOVS” to “ANOVA”.
Reply:
Thank you for your attention to detail. The article has been thoroughly proofread, and minor corrections such as "ANOVS" have been rectified to "ANOVA."

Reviewer 2 Report
Comments and Suggestions for Authors
The manuscript presents results after nasal administration of Apelin-13 in a model of cognitive deficit induced by streptozotocin. Some points must be corrected:
a) The introduction should contain some general data about Apelins and recently reported actions in the Central nervous system.
b) Discussion requires multiple support of analysis of results and relationship with recent works in the field. As examples:
Chen, B., Wu, J., Hu, S., Liu, Q., Yang, H., & You, Y. (2023). Apelin-13 improves cognitive impairment and repairs hippocampal neuronal damage by activating PGC-1α/PPARγ signaling. Neurochemical Research, 48(5), 1504-1515.
KamiÅ„ska, K., Borzuta, H., Buczma, K., & Cudnoch-JÄ™drzejewska, A. (2024). Neuroprotective effect of apelin-13 and other apelin forms—a review. Pharmacological Reports, 1-13.
Azhir, M., Gazmeh, S., Elyasi, L., Jahanshahi, M., & Bazrafshan, B. (2023). The effect of apelin-13 on memory of scopolamine-treated rats and accumulation of amyloid-beta plaques in the hippocampus. Journal of Clinical and Basic Research, 7(4), 15-19.
c) Also, discussion should be clear about the scope of results supporting the role of Nrf2-OH-1 system in the observed results. And suggest clearly which other systems could be involved and studied in short-term.
d) Limitations of the study and specific prospective studies could be sentenced in the end of discussion. Some sentences as "Further research aimed at elucidating the precise molecular mechanisms involved and evaluating the efficacy of Nrf2-HO1 pathway modulation in preclinical and clinical settings is warranted" seem unspecific and speculative.
e) Finally, conclusions section is desirable. The end of discussion section is not clear about this.
Comments on the Quality of English Language
The manuscript contains some grammar mistakes. Edition particularly on discussion section could improve presentation and sense of discussion. In the current form some sentences seem speculative or not clearly supported by results.
Author Response
Reviewers' comments: 
Reviewer #2:
Comments and Suggestions for Authors
The manuscript presents results after nasal administration of Apelin-13 in a model of cognitive deficit induced by streptozotocin. Some points must be corrected:
Comments:
- a) The introduction should contain some general data about Apelins and recently reported actions in the Central nervous system.
Reply:
Thank you for your suggestion. The introduction has been revised to include general information about Apelins and their reported actions in the central nervous system. This addition aims to provide readers with a broader context for understanding the potential therapeutic effects of Apelin-13 in the context of Alzheimer's disease.
Comments:
- b) Discussion requires multiple support of analysis of results and relationship with recent works in the field. As examples:
Chen, B., Wu, J., Hu, S., Liu, Q., Yang, H., & You, Y. (2023). Apelin-13 improves cognitive impairment and repairs hippocampal neuronal damage by activating PGC-1α/PPARγ signaling. Neurochemical Research, 48(5), 1504-1515.
KamiÅ„ska, K., Borzuta, H., Buczma, K., & Cudnoch-JÄ™drzejewska, A. (2024). Neuroprotective effect of apelin-13 and other apelin forms—a review. Pharmacological Reports, 1-13.
Azhir, M., Gazmeh, S., Elyasi, L., Jahanshahi, M., & Bazrafshan, B. (2023). The effect of apelin-13 on memory of scopolamine-treated rats and accumulation of amyloid-beta plaques in the hippocampus. Journal of Clinical and Basic Research, 7(4), 15-19.
Reply:
Thank you for your recommendation. The discussion section has been updated to include analysis and integration of our results with recent works in the field, including the studies you provided.
Comments:
- c) Also, discussion should be clear about the scope of results supporting the role of Nrf2-OH-1 system in the observed results. And suggest clearly which other systems could be involved and studied in short-term.
Reply:
Thank you for your suggestions. We have revised the discussion section to provide clarity regarding the scope of results supporting the role of the Nrf2-HO1 system in our study and suggest clearly which other systems could be involved and studied in short-term.
Comments:
- d) Limitations of the study and specific prospective studies could be sentenced in the end of discussion. Some sentences as "Further research aimed at elucidating the precise molecular mechanisms involved and evaluating the efficacy of Nrf2-HO1 pathway modulation in preclinical and clinical settings is warranted" seem unspecific and speculative.
Reply:
Thank you for your suggestions. We have revised the discussion section. And we also feel that it is also easy to produce confusion for authors, so we have now removed this sentence in the revised version.
Comments:
- e) Finally, conclusions section is desirable. The end of discussion section is not clear about this.
Reply:
Thank you for your comments. We have revised the end of discussion section as follow:
“Initially, we explored the protective mechanism of the intranasal administration of Apelin-13 against STZ-induced nerve injury. Importantly, it was confirmed for the first time that the administration of Apelin-13 through the nose can produce effective drug effects, similar to the effects of previous intraventricular administration techniques. Overall, these behavioral findings provide important insights into the mechanisms underlying the cognitive-enhancing effects of Apelin-13 and underscore the therapeutic potential of targeting the Nrf2-HO1 pathway for the treatment of Alzheimer's disease. The study has several limitations that need to be considered. First, the AD model used in this study is relatively homogeneous, and further investigation is needed to confirm whether similar mechanisms operate in other animal models of AD. Second, the up-stream and downstream molecules regulated by apelin-13 in the Nrf2-HO1 pathway need to be explored in depth to provide a stronger basis for target development. These limitations highlight areas for future research aimed at expanding our understanding of the therapeutic potential of apelin-13 in AD.”
Comments on the Quality of English Language
Comments:
The manuscript contains some grammar mistakes. Edition particularly on discussion section could improve presentation and sense of discussion. In the current form some sentences seem speculative or not clearly supported by results.
Reply:
Thank you for your feedback. We have carefully reviewed the discussion section and made the necessary edits to improve grammar and clarity. Additionally, we have revised sentences that may have seemed speculative or not adequately supported by results. Our goal is to ensure that the discussion section presents a coherent and well-supported interpretation of the findings, avoiding any speculative language. We appreciate your attention to detail and constructive suggestions for enhancing the quality of the manuscript.

Reviewer 3 Report
Comments and Suggestions for Authors
The paper is written correctly and can be easily understood. The topic is attractive and interesting, thus will probably attract a considerable scientific attention. The aim of the paper is to reveal the mechanism of Apelin-13 action via Nrf2-HO1 signaling pathways on a cognitive improvement in Alzheimer's disease (AD) patients. However, the paper has some disadvantages which could be improved to increase the quality of this study. In accordance, I suggest the following corrections:
1. Introduction, line 59: “While oxidative stress is recognized as a crucial mechanism underlying Apelin-13's 59 anti-Alzheimer's disease (AD) effects,…” – a reference (references) should be added.
2. Introduction, the last paragraph: “technical methodologies” – it seems this term is not adequate. Please revise and correct it.
3. Introduction: I suggest the authors to merge and reorganize the last 2 paragraph. The last paragraph should mention the parameters that were analyzed (and their indications), in vivo model system used, and donepezil as a positive control (standard AD therapeutic). Then, the clear aim of the paper, novelty, and scientific significance of the paper compared to literature should be highlighted.
4. Materials and methods, Drug administration: Include donepezil administration in this paragraph.
5. Materials and methods: Add the paragraph “experimental protocol”, which will contain data on the studied experimental groups (agent and dose of treatment, number of animals, etc.), time course of the experiment and other relevant information. Provide an appropriate reference for streptozotocin-induced AD model.
6. Materials and methods, In vitro electrophysiology: “LTP” – define this abbreviation.
7. Materials and methods, The detection of oxidative stress levels: “groups C, L, M, and H” – define these groups in “experimental protocol”. Then, “SOD, MDA, GSH-Px, and CAT levels” – provide full names of these abbreviations. As I understood, the authors determined the (specific) activities of the antioxidant enzymes and the level (concentration) of MDA - please correct this.
8. Materials and methods, Statistical analysis: p < 0.05 – use the small “p” throughout the whole text and figure captions (not P).
9. Results, Intranasal administration of Apelin-13 reduces the oxidative stress of the hippocampal in STZ-induced animal models of AD mice: “hippocampal” – correct this to “hippocampal tissue or hippocampus”. Then, correct to “MDA level” (not the activity of MDA). The authors abbreviated “glutathione-peroxidase” above as GSH-Px (not as “GSH” – abbreviation for “glutathione”) – correct it. Then, “which was consistent with previous reports” – add appropriate references.
10. Results, Figure 1: The caption of Figure 1 is provided in the form of “message” (generally for all figure captions), not as the description (explanation). Please revise all figure captions. Then, “high apelin, low apelin” are metioned in the figure (s). The values of high and low doses of apelin should be provided (in the experimental protocol and/or figure captions), as well as for donepezil dose. Y-axis captions are poorly visible.
11. Results, Figure 3: “Quantitative mean U/mg…” – correct to “specific activity (U/mg) of…”.
12. Discussion: I suggest reorganizing the last 2 paragraphs as effective and clear conclusions of the paper.
13. References: Since AD is a widely-studied topic from different aspects, I am sure that there are numerous relevant references that could be used for this paper.
Author Response
Reviewer #3:
Comments and Suggestions for Authors
The paper is written correctly and can be easily understood. The topic is attractive and interesting, thus will probably attract a considerable scientific attention. The aim of the paper is to reveal the mechanism of Apelin-13 action via Nrf2-HO1 signaling pathways on a cognitive improvement in Alzheimer's disease (AD) patients. However, the paper has some disadvantages which could be improved to increase the quality of this study. In accordance, I suggest the following corrections:
Comments:
- Introduction, line 59:“While oxidative stress is recognized as a crucial mechanism underlying Apelin-13's 59 anti-Alzheimer's disease (AD) effects,…” – a reference (references) should be added.
Reply:
Thank you for bringing this to our attention. We have now added the necessary reference to support the statement regarding oxidative stress as a crucial mechanism underlying Apelin-13's anti-Alzheimer's disease effects in the introduction. This ensures that the claim is properly supported by existing literature.
Comments:
- Introduction, the last paragraph: “technical methodologies” – it seems this term is not adequate. Please revise and correct it.
Reply:
Thank you for your suggestions. We appreciate your suggestion regarding the term "technical methodologies" in the last paragraph of the introduction. We agree that the term could be refined for clarity. We will revise it to methodological approaches to accurately convey the methods utilized in the study.
Comments:
- Introduction: I suggest the authors to merge and reorganize the last 2 paragraph. The last paragraph should mention the parameters that were analyzed (and their indications), in vivomodel system used, and donepezil as a positive control (standard AD therapeutic). Then, the clear aim of the paper, novelty, and scientific significance of the paper compared to literature should be highlighted.
Reply:
Thank you for your suggestions. We have revised the last two paragraphs in the revised version as follows:
“While oxidative stress is recognized as a crucial mechanism underlying Apelin-13's anti-AD effects, the specific oxidative stress pathway through which apelin mitigates AD-related neurological damage requires further investigation. To assess the effective-ness of Apelin-13, we employed STZ-induced AD mice as an in vivo model system and included donepezil, a standard AD therapeutic, as a positive control. By analyzing parameters such as cognitive function, synaptic plasticity, oxidative stress markers, and signaling pathways, we aimed to clarify the therapeutic potential of Apelin-13 and its mechanism of action. Employing a range of methodological approaches, and our study aims to elucidate the precise molecular pathways through which Apelin-13 exerts its neuroprotective effects against AD-associated oxidative stress. The novelty and scientific significance of our study lie in the exploration of intranasal Apelin-13 administration as a potential therapeutic strategy for AD, offering insights into its neuroprotective effects and underlying mechanisms.”
Comments:
- Materials and methods, Drug administration: Include donepezil administration in this paragraph.
Reply:
Thank you for your comments. We have added the donepezil administration in the drug administration.
Comments:
- Materials and methods: Add the paragraph “experimental protocol”, which will contain data on the studied experimental groups (agent and dose of treatment, number of animals, etc.), time course of the experiment and other relevant information. Provide an appropriate reference for streptozotocin-induced AD model.
Reply:
Thank you for your suggestions. We have added the paragraph “experimental protocol” in Materials and methods.
Comments:
- Materials and methods, In vitroelectrophysiology: “LTP” – define this abbreviation.
Reply:
Thank you for your comments. We have defined the LTP in Materials and methods.
Comments:
- Materials and methods, The detection of oxidative stress levels:“groups C, L, M, and H” – define these groups in “experimental protocol”. Then, “SOD, MDA, GSH-Px, and CAT levels” – provide full names of these abbreviations. As I understood, the authors determined the (specific) activities of the antioxidant enzymes and the level (concentration) of MDA - please correct this.
Reply:
Thank you for your valuable comments. We have redefined the subgroups as 1 to 5 groups, clearly outlining the specific treatments each group received in the experimental protocol section. Additionally, we've provided the full names of the abbreviations for the detection of oxidative stress levels and adjusted the descriptions accordingly. We appreciate your professional advice and are grateful for the opportunity to improve the clarity and accuracy of our work.
Comments:
- Materials and methods, Statistical analysis: p < 0.05 – use the small “p” throughout the whole text and figure captions (not P).
Reply:
Certainly, we have ensured to consistently use the small “p” throughout the entire text and figure captions, as per your suggestion. Thank you for bringing this to our attention.
Comments:
- Results, Intranasal administration of Apelin-13 reduces the oxidative stress of the hippocampal in STZ-induced animal models of AD mice: “hippocampal” – correct this to “hippocampal tissue or hippocampus”. Then, correct to “MDA level” (not the activity of MDA). The authors abbreviated “glutathione-peroxidase” above as GSH-Px (not as “GSH” – abbreviation for “glutathione”) – correct it. Then, “which was consistent with previous reports” – add appropriate references.
Reply:
Thank you for your suggestions. We have made the necessary corrections in the Results section. Instead of "hippocampal," we used " hippocampus" for clarity. Additionally, we corrected "the level of MDA" to accurately represent the measurement. The abbreviation for "glutathione-peroxidase" was corrected to GSH-Px, as previously indicated. Regarding the statement "which was consistent with previous reports," we included appropriate references to support this claim. Thank you for your attention to detail.
Comments:
- Results, Figure 1: The caption of Figure 1 is provided in the form of “message” (generally for all figure captions), not as the description (explanation). Please revise all figure captions. Then, “high apelin, low apelin” are metioned in the figure (s). The values of high and low doses of apelin should be provided (in the experimental protocol and/or figure captions), as well as for donepezil dose. Y-axis captions are poorly visible.
Reply:
Thank you for your suggestions. Figure 1 and 2 captions have been revised to provide descriptions instead of messages, enhancing clarity for readers. The values for the high and low doses of apelin, as well as the donepezil dose, have been included in the experimental protocol. Additionally, adjustments have been made to ensure that the y-axis captions are more visible.
Comments:
- Results, Figure 3: “Quantitative mean U/mg…” – correct to “specific activity (U/mg) of…”.
Reply:
Thank you for your suggestions. We corrected “Quantitative mean U/mg…” to the “specific activity (U/mg) of…”.
Comments:
- Discussion: I suggest reorganizing the last 2 paragraphs as effective and clear conclusions of the paper.
Reply:
Thank you for your suggestions. We have revised the last two paragraphs in the revised version as follows:
“While oxidative stress is recognized as a crucial mechanism underlying Apelin-13's anti-AD effects, the specific oxidative stress pathway through which apelin mitigates AD-related neurological damage requires further investigation. To assess the effective-ness of Apelin-13, we employed STZ-induced AD mice as an in vivo model system and included donepezil, a standard AD therapeutic, as a positive control. By analyzing parameters such as cognitive function, synaptic plasticity, oxidative stress markers, and signaling pathways, we aimed to clarify the therapeutic potential of Apelin-13 and its mechanism of action. Employing a range of methodological approaches, and our study aims to elucidate the precise molecular pathways through which Apelin-13 exerts its neuroprotective effects against AD-associated oxidative stress. The novelty and scientific significance of our study lie in the exploration of intranasal Apelin-13 administration as a potential therapeutic strategy for AD, offering insights into its neuroprotective effects and underlying mechanisms.”
Comments:
- References: Since AD is a widely-studied topic from different aspects, I am sure that there are numerous relevant references that could be used for this paper.
Reply:
Thank you for your feedback. We have carefully reviewed our references and will ensure to include a comprehensive selection of relevant literature on Alzheimer's disease from various perspectives to enhance the scientific robustness of our paper.

Round 2
Reviewer 1 Report
Comments and Suggestions for Authors
My comments were addressed and the materials and methods section was significantly improved.
I recommend acceptance of the manuscript.